# The Epidemiology of Obstructive Sleep Apnea in Poland—Polysomnography and Positive Airway Pressure Therapy

**DOI:** 10.3390/ijerph18042109

**Published:** 2021-02-22

**Authors:** Wojciech Kuczyński, Aleksandra Kudrycka, Aleksandra Małolepsza, Urszula Karwowska, Piotr Białasiewicz, Adam Białas

**Affiliations:** 1Department of Sleep Medicine and Metabolic Disorders, Medical University of Lodz, 92-215 Lodz, Poland; aleksandra.kudrycka@stud.umed.lodz.pl (A.K.); aleksandra.malolepsza1@stud.umed.lodz.pl (A.M.); urszula.karwowska@stud.umed.lodz.pl (U.K.); piotr.bialasiewicz@umed.lodz.pl (P.B.); 2Department of Pathobiology of Respiratory Diseases, Medical University of Lodz, 90-153 Lodz, Poland; adam.bialas@umed.lodz.pl

**Keywords:** epidemiology, obstructive sleep apnea syndrome, OSAS, polysomnography, PSG

## Abstract

The aim of this study is to provide a brief summary of the epidemiological data on obstructive sleep apnea syndrome (OSAS) diagnosis and therapy in different regions of Poland from 2010 to 2019. We performed a retrospective study in the sleep center of the Department of Sleep Medicine and Metabolic Disorders, Medical University of Lodz, Poland. We requested data from the National Health Service concerning the number of new diagnoses of OSAS, the polysomnographies (PSGs) that were performed, and reimbursements of positive airway pressure (PAP) therapy in each region of Poland in the period 2010–2019. The constant increase in the number of polysomnographies performed and PAP reimbursements suggests the need to create a national network between regional sleep centers to provide proper care for patients with OSAS, and PAP therapy.

## 1. Introduction

Obstructive sleep apnea syndrome (OSAS) is characterized by repeated episodes of partial or complete breathing cessation during sleep due to pharyngeal airway closure [1]. Common symptoms of OSAS include excessive daytime sleepiness, loud snoring, recurrent arousals during sleep, as well as morning headaches [2]. The prevalence of OSAS in the adult population is estimated to be 3–7% [3]; however, some recent studies suggest that OSAS is considerably more frequent and affects up to approximately 84% of men and 61% of women [4,5]. Pływaczewski et al. estimated the prevalence of obstructive sleep apnea syndrome in Poland at 7.5% on the basis of a group of 676 patients from Warsaw [6]. To our knowledge, information about general epidemiological data on OSAS diagnosis in Poland is limited. We wish to highlight that this is the first OSAS epidemiological study performed in Poland.

To the major risk factors for OSAS we can include obesity, hypertension, male sex, and age. Moreover, OSAS leads to the development of many severe health consequences, such as cardiovascular, cerebrovascular, as well as endocrine and metabolic disorders [2,7,8,9,10,11]. Furthermore, OSAS is related to impairment in work performance and a higher risk of occupational and industrial accidents [12,13]. The number of possible consequences of OSAS shows the importance of both proper diagnosis and adequate treatment.

The American Academy of Sleep Medicine (AASM) defined four types of sleep study devices used for diagnostic testing for sleep disorders [14,15,16]. A Type I study, which is considered as the “gold standard” in OSAS diagnosis, entails polysomnography (PSG). This is an attended, full laboratory examination, which uses at least 7 monitoring channels: electroencephalography, electrocardiography, electromyography, electrooculography, airflow, oxygen saturation, and respiratory effort. However, there are several limitations of PSG, such as low accessibility, high cost, and requirement of highly trained technologists for data collection and interpretation. A Type II study constitutes an unattended, full polysomnography (with at least 7 channels); therefore, it does not require access to a sleep laboratory. A Type III study entails modified, portable apnea testing, which measures at least four parameters: oxygen saturation, two respiratory variables (respiratory movement and airflow), and a cardiac variable (heart rate or electrocardiogram). A Type IV study measures only one or two parameters, usually oxygen saturation or airflow. The OSAS diagnosis is based on the apnea–hypopnea index (AHI) during nocturnal polysomnography. The AHI is defined as the number of apneas and hypopneas per hour of sleep and represents the OSAS severity: mild (AHI of 5–15 events/h), moderate (AHI of 15–30 events/h), and severe (AHI > 30 events/h) [17,18,19].

The AASM propose several options for OSAS treatment in adult patients: positive airway pressure (PAP) treatment options; oxygen therapy; oral appliance therapy; surgical treatment options; hypoglossal nerve stimulators; nasal resistive valves; and pharmacological therapies. Due to the significant body of evidence supporting its impact on clinical outcomes, PAP is considered as a first-line therapy for the management of OSAS [14]. The main limitation to the efficacy of PAP therapy is the willingness of patients to accept PAP therapy up front and, in those who do, remaining adherent to therapy over time. According to the current authors’ knowledge, there are no national data regarding compliance of PAP therapy in the Polish population. 

Initially, we performed a retrospective study in the sleep center of the Department of Sleep Medicine and Metabolic Disorders, Medical University of Lodz, Poland. However, we realized that there is neither national nor regional data about the general epidemiology of OSAS in Poland. Therefore, we requested data from the National Health Service to further study this field. We hope that a deeper understanding of this issue will indicate some specific needs and clarify problems associated with diagnostic incidence and treatment of OSAS in Poland, providing material for further discussion and improvement of sleep-related healthcare by the development of dedicated systemic solutions.

The aim of this study is to provide a brief summary of the epidemiological data on the incidence of OSAS diagnosis and prescribed therapy in different regions of Poland from 2010 to 2019. 

## 2. Materials and Methods

We requested raw epidemiological data from the National Health Service to answer the following questions: How many new diagnoses of OSAS have been made during 2010–2019 in each region of Poland?How many PSGs have been performed during 2010–2019 in each region of Poland?How many reimbursements of PAP therapy have occurred during 2010–2019 in each region of Poland?

The National Health Service in Poland gathers statistical data exclusively on Type I devices; other types of devices (Type II–IV) were excluded from the study. A PSG performed as a nocturnal PAP titration was excluded from the study as well.

The data regarding number of habitants in each region of Poland were obtained from the Central Statistical Office for 2019 and we assumed no significant changes among number of habitants in each of the regions during the previous 10 years. Frequencies in the separate regions were compared using multi-way tables and chi^2^ tests.

## 3. Results

All variables of interest are listed in Table 1 and Figure 1. The total number of new diagnoses of OSAS, polysomnographies, and reimbursements were calculated per 100,000 habitants in each of the regions of Poland. The prevalence of OSAS diagnosis per 100,000 habitants is highest in the regions Kujawsko-Pomorskie (*N* = 1328), Świętokrzyskie (*N* = 1028), and Mazowieckie (*N* = 918). A PSG was performed the most frequently in the regions Świętokrzyskie (*N* = 910), Lubuskie (*N* = 803), and Opolskie (*N* = 803). PAP therapy was the most common in the regions Kujawsko-Pomorskie (*N* = 265), Świętokrzyskie (*N* = 257), and Lubuskie (*N* = 244).

We observed statistically significant differences among regions according to the number of new diagnoses of OSAS (chi^2^ = 22,052, *p* < 0.001), polysomnographies (chi^2^ = 19,242, *p* < 0.001), and reimbursements (chi^2^ = 5631, *p* < 0.001).

The diagnosis of OSAS assumes an AHI of >5 events/h. The data do not indicate the severity of the disease.

## 4. Discussion

Our data provide a general overlook of the epidemiology of diagnosis and PAP therapy of OSAS in Poland from 2010 to 2019. Our data suggest that, depending on the region, OSAS diagnoses range from 390 to 1328/100,000 habitants (*p* < 0.001). As expected, the regions differ significantly according to the number of new diagnoses of OSAS, polysomnographies, and PAP reimbursements. The economic, social, and demographic differences will be investigated in additional studies. To the author’s knowledge, this is the first study that summarizes the epidemiology of obstructive sleep apnea in Poland and provides a general overlook of the epidemiological data.

The data highlight that, in each region of Poland, the number of performed PSGs increases continuously with every year, as presented in Figure 1D,E. This trend is followed by an increase in the number of newly diagnosed OSAS cases. The number of reimbursements of PAP therapy is lower than the number of performed PSGs and diagnoses of OSAS. However, this trend was not observed for the Kujawsko-Pomorskie, Lubuskie, Łódzkie, Małopolskie, Mazowieckie, and Śląskie regions, where the increase in the number of PAP reimbursements is accompanied by a decrease in the number of polysomnographies performed. One of the reasons that can be discussed in this context is an increasing number of polygraphies (PGs), which are often performed instead of an PSG. Indeed, PG is one of the most available tools to screen for OSAS, but is limited by the lack of professional training for physicians to perform and score the PG results, which may lead to diagnostic pitfalls.

Another important issue is that, in Poland, there are just a few sleep centers specialized in sleep-related disorders other than sleep-related breathing disorders, which may lead to limited comprehensiveness of the diagnostic process. 

There are no data on compliance with PAP usage at home. Non-invasive mechanical ventilation (NIV) is an important player in this field. Implementation of the “National program to reduce mortality from chronic respiratory diseases by creating NIV units in the years 2016–2019” (acronym: POL-VENT) provided numerous departments of general pulmonology with PSG and PG equipment. This fact may explain a continuous increase in the number of PSGs performed, as well as the number of new patients with an OSAS diagnosis. The constant increases in the number of polysomnographies performed and PAP reimbursements suggest the need to create a national network between regional sleep centers to provide proper care for patients with OSAS, and PAP therapy. It should be discussed whether there are conditions to propose a new medical specialization in sleep medicine, as what the AASM or European Sleep Research Society (ESRS) provides, that would emphasize the need for providing diagnoses and therapy to conditions other than sleep-related breathing disorders. Undoubtedly, obtaining an international, European-level certification would be beneficial for increasing the knowledge and standards of sleep medicine in Poland. Such an opportunity is offered by the ESRS in the form of an examination in sleep medicine, the passing of which results in the title of somnologist—an expert in sleep medicine. Currently, the database of this society shows that there is only one certified somnologist in Poland. It also seems reasonable to focus on creating national guidelines on sleep medicine, especially on sleep diagnostic procedures, to regulate, among others, the usage of PG. 

Our study has some limitations. First of all, the data provided in this summary are based on data available from the National Health Service and do not refer to particular sleep centers. There also are no data on diagnostic types other than PSG, distinguished by the AASM to make a diagnosis of OSAS. Moreover, the data do not cover the private sector, which is a potentially significant player in this field. Furthermore, as we previously mentioned, the data do not provide detail on the severity of the disease.

## 5. Conclusions

To summarize, the data in this study underestimate the epidemiology of OSAS in Poland but illustrate the overall landscape of this issue and open the debate in this field.

## Figures and Tables

**Figure 1 ijerph-18-02109-f001:**
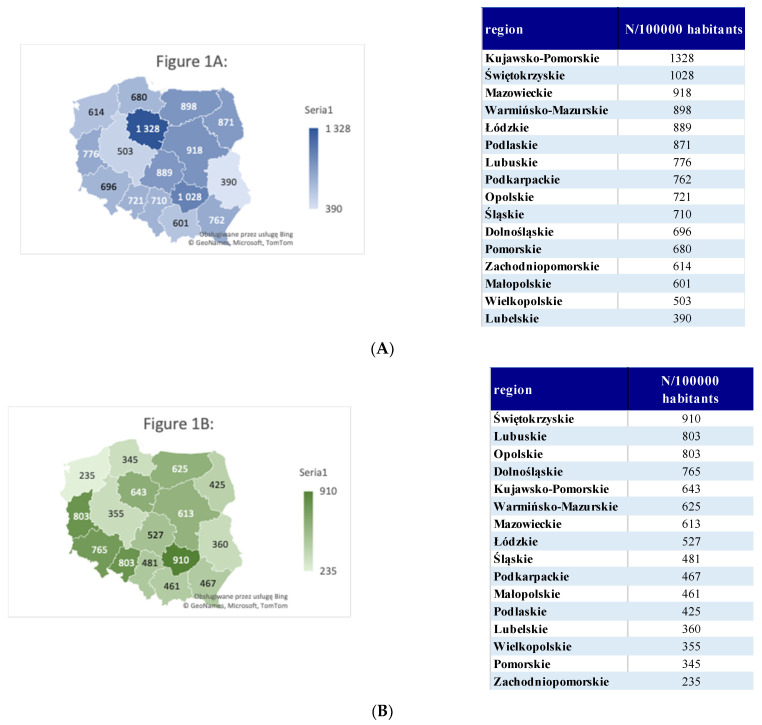
Number of patients with a diagnosis of obstructive sleep apnea syndrome (OSAS), the number of polysomnographies, and the number of reimbursements of positive airway pressure (PAP) treatment in regions of Poland from 2010 to 2019. (**A**) Number of patients with a diagnosis of obstructive sleep apnea syndrome during 2010–2019 in all regions of Poland. (**B**) Number of polysomnographies during 2010–2019 in all regions of Poland. (**C**) Number of reimbursements by the National Health Service for the positive airway pressure (PAP) therapy for obstructive sleep apnea. (**D**,**E**) The relationship and the trends between the number of patients with an OSAS diagnosis, the PSG performed, and PAP reimbursements for the years 2010–2019. * 2019 from January to June.

**Table 1 ijerph-18-02109-t001:** Number of patients with a diagnosis of obstructive sleep apnea syndrome (OSAS), the number of polysomnographies, and the number of reimbursements of positive airway pressure (PAP) treatment in all regions of Poland from 2010 to 2019. The bold in number is summary of all data provided.

Part 1. Number of Patients with a Diagnosis of Obstructive Sleep Apnea Syndrome (OSAS) during 2010–2019 in All Regions of Poland.													
Region	Year												
	2010	2011	2012	2013	2014	2015	2016	2017	2018	2019 *	Total	Number of Habitants	N/100,000 Habitants
**Dolnośląskie**	1264	1502	1694	1670	1807	2087	2489	2825	3031	1819	**20,188**	**2,901,225**	**696**
**Kujawsko-Pomorskie **	2063	2126	2191	2457	2437	2776	3224	3702	3942	2678	**27,596**	**2,077,775**	**1328**
**Lubelskie **	529	606	719	823	463	589	1001	1199	1374	961	**8264**	**2,117,619**	**390**
**Lubuskie**	367	434	549	760	369	734	908	1202	1448	1105	**7876**	**1,014,548**	**776**
**Łódzkie**	1478	1579	1695	2040	2024	2337	2649	2828	3124	2170	**21,924**	**2,466,322**	**889**
**Małopolskie **	1336	1304	1379	1418	1455	1841	2392	3240	3679	2390	**20,434**	**3,400,577**	**601**
**Mazowieckie**	3133	3817	4388	4905	4996	5343	5478	6137	6776	4618	**49,591**	**5,403,412**	**918**
**Opolskie **	244	357	462	637	670	943	885	960	1188	771	**7117**	**986,506**	**721**
**Podkarpackie**	765	910	1209	1578	1693	1790	1968	2231	2471	1617	**16,232**	**2,129,015**	**762**
**Podlaskie**	744	747	766	767	794	1061	1346	1480	1640	947	**10,292**	**1,181,533**	**871**
**Pomorskie **	1328	1245	959	1193	1313	1582	1939	2318	2436	1565	**15,878**	**2,333,523**	**680**
**Śląskie**	2545	2934	3098	3322	2602	3024	3567	3677	4463	2943	**32,175**	**4,533,565**	**710**
**Świętokrzyskie**	251	745	943	1192	1490	1409	1793	1900	1850	1194	**12,767**	**1,241,546**	**1028**
**Warmińsko-Mazurskie**	527	847	868	939	1081	1219	1807	1995	2153	1399	**12,835**	**1,428,983**	**898**
**Wielkopolskie **	1072	1123	1220	1362	1646	1910	2223	2587	2907	1535	**17,585**	**3,493,969**	**503**
**Zachodniopomorskie**	341	474	593	992	1272	1367	1494	1526	1464	929	**10,452**	**1,701,030**	**614**
**Poland Summary**	**17,987**	**20,750**	**22,733**	**26,055**	**26,112**	**30,012**	**35,163**	**39,807**	**43,946**	**28,641**	**291,206**	**38,411,148**	**758**
*** 2019 from January to June**													
**Part 2. Number of Polysomnographies during 2010–2019 in All Regions of Poland.**													
**Region**	**Year**												
	**2010**	**2011**	**2012**	**2013**	**2014**	**2015**	**2016**	**2017**	**2018**	**2019 ***	**Total**	**Number of Habitants**	**N/100000 Habitants**
**Dolnośląskie**	1650	3071	2387	2541	1640	1638	2111	2419	3138	1595	**22,190**	**2,901,225**	**765**
**Kujawsko-Pomorskie **	1842	1744	1624	1427	907	1059	1212	1375	1359	814	**13,363**	**2,077,775**	**643**
**Lubelskie **	663	668	856	926	413	591	852	920	1067	661	**7617**	**2,117,619**	**360**
**Lubuskie**	528	584	655	899	490	838	1003	1154	1236	758	**8145**	**1,014,548**	**803**
**Łódzkie**	1156	1093	1118	1215	1300	1294	1693	1610	1648	881	**13,008**	**2,466,322**	**527**
**Małopolskie **	1584	1668	1430	1430	1055	1287	1548	2084	2221	1372	**15,679**	**3,400,577**	**461**
**Mazowieckie**	2614	3423	3613	3480	3617	3718	3287	3481	3721	2165	**33,119**	**5,403,412**	**613**
**Opolskie **	284	424	561	818	766	1174	946	1013	1144	789	**7919**	**986,506**	**803**
**Podkarpackie**	643	769	1052	1378	1296	1071	1008	1142	987	596	**9942**	**2,129,015**	**467**
**Podlaskie**	540	589	581	744	470	369	399	478	542	307	**5019**	**1,181,533**	**425**
**Pomorskie **	1126	414	503	593	655	866	958	1096	1193	648	**8052**	**2,333,523**	**345**
**Śląskie**	2581	2631	2551	2639	1494	1986	2142	2076	2467	1260	**21,827**	**4,533,565**	**481**
**Świętokrzyskie**	265	717	800	995	1248	1290	1811	1800	1598	778	**11,302**	**1,241,546**	**910**
**Warmińsko-Mazurskie**	460	630	634	693	788	878	1423	1474	1366	583	**8929**	**1,428,983**	**625**
**Wielkopolskie **	958	988	1038	1103	1346	1367	1507	1701	1673	723	**12,404**	**3,493,969**	**355**
**Zachodniopomorskie**	169	185	415	597	711	622	399	369	345	190	**4002**	**1,701,030**	**235**
**Poland Summary**	**17,063**	**19,598**	**19,818**	**21,478**	**18,196**	**20,048**	**22,299**	**24,192**	**25,705**	**14,120**	**202,517**	**38,411,148**	**527**
*** 2019 from January to June**													
**Part 3. Number of Reimbursements by the National Health Service of Positive Airway Pressure (PAP) in the Therapy of Obstructive Sleep Apnea Syndrome (OSAS).**													
**Region**	**Year**												
	**2010**	**2011**	**2012**	**2013**	**2014**	**2015**	**2016**	**2017**	**2018**	**2019 ***	**Total**	**Number of Habitants**	**N/100,000 Habitants**
**Dolnośląskie**	269	334	432	379	373	466	667	841	1019	1330	**6110**	**2,901,225**	**211**
**Kujawsko-Pomorskie **	257	407	456	397	446	423	480	626	794	1222	**5508**	**2,077,775**	**265**
**Lubelskie **	92	96	140	161	166	231	298	337	479	504	**2504**	**2,117,619**	**118**
**Lubuskie**	115	112	125	149	161	233	279	368	397	533	**2472**	**1,014,548**	**244**
**Łódzkie**	220	256	298	230	321	477	655	750	790	970	**4967**	**2,466,322**	**201**
**Małopolskie **	255	243	280	319	479	565	809	1222	1468	1714	**7354**	**3,400,577**	**216**
**Mazowieckie**	492	637	643	723	857	996	1249	1541	1878	2212	**11,228**	**5,403,412**	**208**
**Opolskie **	34	58	88	116	83	114	115	155	274	258	**1295**	**986,506**	**131**
**Podkarpackie**	50	137	273	273	299	394	414	584	634	689	**3747**	**2,129,015**	**176**
**Podlaskie**	35	42	48	72	89	109	108	166	212	271	**1152**	**1,181,533**	**98**
**Pomorskie **	199	178	149	177	179	273	330	492	543	631	**3151**	**2,333,523**	**135**
**Śląskie**	373	369	374	424	420	588	755	884	1091	1396	**6674**	**4,533,565**	**147**
**Świętokrzyskie**	108	153	198	247	362	369	395	398	444	490	**3164**	**1,241,546**	**255**
**Warmińsko-Mazurskie**	122	149	136	136	149	209	282	461	471	475	**2590**	**1,428,983**	**181**
**Wielkopolskie **	177	192	206	259	338	346	507	572	744	674	**4015**	**3,493,969**	**115**
**Zachodniopomorskie**	43	56	75	110	153	143	115	154	196	274	**1319**	**1,701,030**	**78**
**Poland Summary**	**2841**	**3419**	**3921**	**4172**	**4875**	**5936**	**7458**	**9551**	**11,434**	**13,643**	**67,250**	**38,411,148**	**175**
*** 2019 from January to June**													

## Data Availability

Data available in a publicly accessible repository that does not issue DOIs Publicly available datasets were analyzed in this study. This data can be found in the National Health Service in Poland.

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
