# Peer review of "The Epidemiology of Obstructive Sleep Apnea in Poland—Polysomnography and Positive Airway Pressure Therapy"

_ijerph, 2021, doi:10.3390/ijerph18042109_

Round 1

Reviewer 1 Report

Although it is clear that  the findings must be confirmed more widely, they represent an interesting stimulus for further study and the paper merits to be published. 

This is a systematic investigation on “ Positive airway pressure therapy pf obstructive sleep apnea in Poland-where we are now? “:

General remarks: This reviewer appreciates the opportunity to evaluate this article with comprehensive data and repeated measurement of variables on this particular topic. From a research standpoint, it is interesting because it seems to support an initial answer for chiarifying  the topic. The impact on  reimbursements of  Osa diagnosis and  CPAP therapy in Poland and is an argument needing more attention for their economical effects also on the various National Health Services in Europe . The suggestion for creating a national network between the sleep centers represents a nice and potential proposal. Although it is clear that the findings must be  confirmed more widely, they represents an interesting stimulus for futher study and   it merits to be published.

Author Response

Thank you very much for your time and effort to review our manuscript. We are very grateful for your valuable comments and understanding for the limitations of our work, honestly described in the text of the manuscript.

Reviewer 2 Report

The manuscript by Dr Kuczyński and colleagues focuses on identification (by National Health Service) of the prevalence and of OSA in a general population in Poland. The authors found an increase in the number of PSGs and PAP prescriptions in different regions. Unfortunately, this manuscript contains only superficial information with many important limitations, already stated by the authors. Furthermore, there is no further statistical analysis showing demographics or comorbidities correlations with OSA diagnosis, some areas of the manuscript are poorly worded, reference list is not updated and discussion section needs improvement.

Author Response

Thank you very much for your time and effort to review our manuscript. We are very grateful for all your valuable comments. As you stated, our report is very superficial, and we honestly described its limitation. However, the character of the data, which is purely based on the reported documentation, determined such kind of only descriptive approach. The study is based on data available from the National Health Service and dose not refer to particular sleep center, therefore, data on comorbidities would be very indirect and hard to verify. Therefore, it would lead to uncertain conclusions. That is why, we focused on prescription data. The point of such study is signalizing the topic to open debate, further discussion and research in the field. The English language and style were reviewed by English native and parts of the manuscript were rewritten. Again, thank you very much for your comments.

Reviewer 3 Report

The authors have worked on a very interesting public health topic. The paper provides an epidemiological summary about obstructive sleep apnea (OSA) diagnosis and therapy in different regions of Poland from 2010 to 2019. Overall, the paper is well written, although some English changes are required. The results of the study are interesting. However, the reviewer would like to point out the following:

  1. Introduction:
    • Acronyms should be previously defined (check Abstract and Introduction).
    • Authors should specify that the apnea-hypopnea index (AHI) is obtained from the patient's physiological recordings and that it is measured in events per hour (e/h). Moreover, they should provide a reference for AHI and severity degree definitions.
    • It should be clarified that the therapies mentioned by the authors are for adult patients.
    • It is not mentioned whether there are previous state-of-the-art studies that have carried out research in this context (background).
    • The objective of the study is clear, but what is the motivation and relevance of the study? How could the results of this study contribute to the improvement of the OSA diagnosis and treatment in the Polish adult population? What value would it have for the scientific community to know these epidemiological data? These study points are very important and should be clarified in “Introduction”.
  2. Materials and Methods:
    • Was AHI ≥ 5 e/h the threshold for determining new diagnoses (subjects with positive OSA) or was it differentiated by severity groups?
    • Why were sleep studies with type 1 devices (polysomnography, PSG) included and not those with type 2, 3 or 4 devices? Sometimes, some subjects undergo PSG several times (before and after treatment to detect the presence of residual OSA). Did the authors take this into account? Do the data presented in the study include/exclude several PSGs performed to the same patient?
  3. Results:
    • Regarding the above, the number of PSGs performed is lower than the number of subjects diagnosed as OSA positive in some regions (for example, 13363 PSGs were performed and 27596 subjects with positive OSA were detected between 2010 and 2019 in Kujawsko-Pomorskie). The data should be consistent. If subjects diagnosed by means of type 1, 2, 3, or 4 devices are included, the number of tests performed using type 2, 3, and 4 devices should also be indicated, and not just with type 1 devices. If this information is not available, the authors should at least indicate the number of subjects diagnosed as OSA positive through PSG and the number of subjects diagnosed by means of another sleep study. This would allow to achieve a more suitable global vision of the relationship between both variables: PSGs performed and subjects diagnosed.
    • In order to support the results of the study, it would be interesting to carry out a statistical analysis to assess whether there are statistically significant differences between the different regions in the number of subjects diagnosed, number of PSGs performed, and number of reimbursements of PAP therapy. These differences could also be evaluated by years.
    • It would also be interesting to can graphically visualize the relationship between the 3 variables by region and/or by year, that is, to be able to see how as one variable increases, the others increase/decrease.

Author Response

Thank you very much for your time and effort to review our manuscript. We are very grateful for all your valuable comments.

We implemented numerous changes to the manuscript according to your comments:

  1. All acronyms are defined. We apologize for this issue that it was not done previously.
  2. The reference ranges for AHI are now appropriately cited
  3. It is clarified that treatment methods are for adult patients
  4. The previous study in the field, which was attempted by the authors, as well as the idea of the research and its possible clinical implication is now mentioned in Introduction
  5. Unfortunately, the diagnosis of OSAS is based on the documentation and assumes AHI >5/h. The data do not provide detailed data on the severity of the disease. We described it both in the results and in limitations sections. In Poland reimbursements from the National Health Service is only for type I PSG, which is considered as a gold standard in the OSAS diagnosis. Other type devices are excluded from the study.

The number of PSG performer referred to diagnostic procedure only without CPAP titration which is a different procedure for the National Health Service and was excluded from the study.

To the author's knowledge, this is the first epidemiological study performed in Poland, results presented in the study show several outcomes which should be implemented by the National Health Service or Ministry of Health to gain better and more visible control on OSAS diagnosis and treatment.

Round 2

Reviewer 2 Report

The manuscript was improved by revision, However, the title could be changed to " Obstructive sleep apnea diagnosis and treatment in Poland – where we are now? 

Author Response

Thank you very much for your time and effort to review our manuscript for the second time. We appreciate Your suggestion to change the title of the manuscript. After a short discussion, we agreed to change it to "The epidemiology of obstructive sleep apnea in Poland – polysomnography and positive airway pressure therapy."

It is worth mentioning that we added a simple statistic as other reviewer suggested. 

We observed statistically significant differences among regions according to a number of new diagnoses of OSAS (chi2=22052, p<0.001), polysomnographies (chi2=19242, p<0.001) and reimbursements (chi2=5631, p<0.001).

Reviewer 3 Report

I appreciate the authors in addressing my concerns and revising the paper with additional information and figures. However, I would like to point out the following:

  • The units of the apnea-hypopnea index are events/h or e/h.
  • The positive airway pressure therapy is sometimes abbreviated as CPAP and other times as PAP. This should be unified.
  • The authors have provided additional information in the “Introduction” of their manuscript. However, this information is superficial in some points of the section and still needs improvement. The authors should deepen further into the problem covered by their study, as well as the benefits and clinical implications that knowing these epidemiological data could have. In order to highlight the novelty of their work, it should be explicitly indicated that this is the first OSAS epidemiological study performed in Poland.
  • In the section “Materials and Methods”, the authors should indicate the cases excluded from the study and its corresponding justification.
  • As I indicated in the first review, it would be interesting to carry out a statistical analysis to assess whether there are statistically significant differences between the different regions of Poland in the number of subjects diagnosed, number of PSGs performed, and number of reimbursements of PAP therapy, in order to support the results of the study.
  • Like the introduction section, the discussion could also be improved.
  • Regarding the English language and style, some changes are still required. In addition, the text should be more fluid and better linked in some sections, such as the introduction or the discussion.

Author Response

Thank you very much for your time and effort to review our manuscript. We are very grateful for your valuable comments and understanding for the limitations of our work, honestly described in the text of the manuscript.

1) 1) First of all, as other reviewer suggested, we decided to change the title of the manuscript "The epidemiology of obstructive sleep apnea in Poland – polysomnography and positive airway pressure therapy" 

2) We unified the manuscript, the units of the apnea-hypopnea index are events/h; as well as PAP instead of CPAP. 

3) We pointed out the exclusion criteria

4) We performed the statistical analysis. Frequencies in the separate regions were compared using multi-way tables and chi2 test. We observed statistically significant differences among regions according to a number of new diagnoses of OSAS (chi2=22052, p<0.001), polysomnographies (chi2=19242, p<0.001) and reimbursements (chi2=5631, p<0.001). The results we received require further investigation to find out the possible correlation with demographic, social, economic characteristics of each region. 

5) The manuscript was verified by the native speaker from the UK who workes in the field of sleep apnea.

This manuscript is a resubmission of an earlier submission. The following is a list of the peer review reports and author responses from that submission.